# The Impact of Sociodemographic Factors on the Rationing of Nursing Care in Urology Wards

Katarzyna Jarosz * and Agnieszka Młynarska

Department of Gerontology and Geriatric Nursing, School of Health Sciences, Medical University of Silesia, 40-055 Katowice, Poland
* Correspondence: katarzyna.jarosz@sum.edu.pl; Tel.: +48-323598191

**Abstract:** Background: The problem of care rationing is widespread all over the world and results from many factors affecting nurses. These factors may result from the environment in which the nurses work, e.g., the atmosphere at work, or may not be related to work, e.g., place of residence. The aim of this study was to examine the impact of sociodemographic factors (place of residence, satisfaction with the financial situation, number of forms of postgraduate education, work system, number of patients per nurse, number of diseases) on care rationing, job satisfaction and quality of nursing care. Methods: The study is a cross-sectional study which includes 130 nurses from all over Poland who work in urology wards. The criteria for inclusion were consent to the examination, practicing the profession of a nurse, work in the urology department and work experience of at least 6 months, regardless of the number of hours worked (full-time/part-time). The study was conducted using the standardized PIRNCA (Perceived Implicit Rationing of Nursing Care) questionnaire. Results: The average rationing nursing care was 1.11/3 points which means nursing care was rarely rationed. The average job satisfaction was 5.95/10 points, and the assessment of the quality of patient care was 6.88/10 points, which means a medium level of the job satisfaction and the quality of patient care. The rationing of care was affected by the number of nurse illnesses; job satisfaction was influenced by the place of residence and satisfaction with the financial situation, while the quality of care was not influenced by any of the analyzed factors. Conclusions: The result of care rationing is at a similar level as the results in Poland and abroad. Despite the rare rationing of care, employers should take corrective action, especially in terms of increasing the staff and health prevention of nurses.

**Keywords:** rationing of nursing care; sociodemographic factors; job satisfaction; quality of care

## 1. Introduction

The work of a nurse in a hospital is specific due to its complexity and multitasking. The nurse is obliged to perform care and treatment tasks, keep records, operate equipment, interview patients about the health condition and many others. In addition, a nurse should support the mentally and spiritually ill [1]. In addition, there are numerous requirements for nurses, such as having a variety of abilities, skills, predispositions, as well as accuracy and precision and vigilance in terms of symptoms reported by patients. In addition, nursing activities should be performed in accordance with detailed procedures, regulations and rules. The work of a nurse has been described as medium–heavy [2], and even as hard and very hard [3]. Nurses, often under the influence of task priorities and inefficient own and organizational factors, are forced to selectively perform activities or omit them, i.e., to ration care [4].

The concept of care rationing was first introduced by nurse Beatrice J. Kalisch in 2006. Subsequently, research on care rationing was undertaken by M. Shubert in 2007 in Switzerland and she created the BERNCA (Basel Extent Rationing of Nursing Care) measurement tool [5]. In 2014, in the USA, this tool was adapted by T. Jones and created the PIRNCA (Perceived Implicit Rationing of Nursing Care) tool [6], and in 2019, the

Polish version was adapted and confirmed with the original by Uchmanowicz et al. [7]. Care rationing occurs when patient care is wholly or partially omitted and when care is required [8]. Therefore, rationing of care can be defined as complete or partial failure to perform necessary nursing activities during the care of a patient [9,10].

Rationing nursing care can be individual and institutional. Institutional rationing can be characterized as a specific operating policy of a given institution that forces employees to take certain actions, e.g., rationing care among nurses. However, the individual is subject to the decision of the person, i.e., the nurse; she does not have any specific norms or rules. A decision is then made on the basis of individual skills, knowledge or ethics. The fact that nurses show less involvement in making decisions about rationing of care may be related to the fact that rationing occurs unconsciously "at the bedside" [11].

This problem is common and present all over the world, and, above all, it can threaten the safety of the patient [5]. Nursing rationing is against the premise of holistic nursing care and leads to a reduction in the quality of services provided by nurses [12]. It occurs mainly when the resources to perform the activities are insufficient [13], which may include a small number of nurses in relation to the number of patients in need of care, modern methods of treatment, and an increasing number of patients whose needs and requirements are ever higher. In addition, care rationing may be influenced by factors related to a given nurse, such as her skills, knowledge and attitudes [14]. The research also showed the influence of other factors closely related to the nurse on the rationing of care, e.g., professional burnout, fatigue, stress, job satisfaction, seniority, age, life satisfaction and life orientation [15–19].

Many studies have been carried out to assess the level of care rationing in wards of various specialties, e.g., intensive care wards [17] and oncology wards [7]. Urology departments tend to be overlooked in many studies or automatically combined with surgical departments. The work of a nurse in the urology ward is focused on perioperative care and detailed preparation of the patient for self-care at home. The process of preparing the patient for self-care begins immediately after admission to the ward and covers the perioperative period, and then consists of instruction on stoma care, diet and individual recommendations tailored to the specific patient [6,20]. In addition, nurses working in urology wards have the closest contact with the intimate sphere of every human being, which additionally puts pressure on them to be able to talk and support the patient, and also requires devoting more time to the patient and caution so as not to violate their dignity.

According to the Report of the Supreme Chamber of Nurses and Midwives, the largest age group are nurses aged 51–60 (36% of all employed nurses), while the distribution of the age structure indicates the lack of replacement by young generations [21]. These results unequivocally indicate that the problem of care rationing in Poland, regardless of the specificity of the work under examination, will be increasing only due to staff shortages, and, additionally, this problem will be aggravated by other previously mentioned factors.

Nurses in the course of their profession and related tasks are affected by various factors that may negatively affect their mental and physical condition [22]. These factors include, among others, atmosphere in the ward, a large number of patients under the care of one nurse, shift work, variety of tasks, technical resources, and work organization [23]. If the risk factors are not eliminated, the quality of care may decrease [24].

Despite much research on care rationing and the factors that influence it, there are still unexplored areas and researchers have not identified which factors are most important and which should be constantly monitored. This particularly applies to Polish healthcare facilities.

Knowledge which factors have impact on rationing nursing care lets to effectively prevent it and ensures patients' safety. Moreover, it can help take the actions that will influence the work places of nurses and additionally increase the efficiency of the nurses' work. The conducted surveys focused on factors closely related to work place. Due to this fact, the authors decided to focus on factors related to their private sphere to check this new space. Therefore, research was undertaken to check the impact of sociodemographic factors on the rationing of care in urology wards, such as postgraduate, workplaces, work

experience in the profession, internship in the urology department, number of diseases, number of the sick, education, operating mode, financial situation, staying on sick leave, willingness to change, marital status, place of residence. Unfortunately, statistical methods included only a few factors mentioned above in the study.

The aim of the study was to assess the impact of sociodemographic factors (place of residence, forms of postgraduate education, work system, number of patients per one nurse on duty, satisfaction with the financial situation, number of diseases the nurse suffers from) on the rationing of nursing care in urology wards.

## 2. Materials and Methods

### 2.1. Study Design

The study is a cross-sectional study. The data were collected for three months, from March to May 2021. The research sample was a group of 130 nurses from all over Poland who work in urology wards and who agreed to participate in the study.

### 2.2. Tools

An anonymous questionnaire consisting of a metric and a standardized PIRNCA questionnaire was used to conduct the study.

The PIRNCA questionnaire was created by Jones in the USA in 2014 [25] based on the BERNCA questionnaire created by Shubert in 2007. The Polish adaptation and translation was performed by Uchmanowicz. The tool is characterized by a high level of reliability and accuracy and its compliance with the original has been confirmed; the Cronbach's alpha was 0.957 [26].

The PIRNCA questionnaire consists of 3 parts. The first consists of 31 statements regarding the rationing of care, the second evaluates the quality of nursing care, and the third concerns job satisfaction. The first part refers to the tasks entrusted in the last seven shifts that could not be performed. It is assessed on a four-point scale from 0 to 3, where 0 means "never", 1—"rarely", 2—"sometimes", 3—"often". If patients under the care of a nurse did not require performing any of the activities, the answer "not applicable" would be marked, and this answer is excluded from the final result. The total rationing score is calculated on the basis of the average of the points (excluding the answers "not applicable") and ranges from 0 to 3, where 0 means "never", 1—"rarely", 2—"sometimes", 3—"often". The next two parts are assessed on a 10-point Likert scale; the higher the score, the higher the job satisfaction or the quality of care [7].

### 2.3. Participants

A total of 130 nurses working in urology departments in Poland participated in the study. The criteria for inclusion were consent to the examination, practicing the profession of a nurse/nurse, work in the urology department and work experience of at least 6 months, regardless of the number of hours worked (full-time/part-time). The exclusion criteria were lack of consent and failure to complete the questionnaire.

Most of the respondents were women (98.5%), and the average age of the respondents was 37.78 years ($\pm$11.86 years; 21–62 years), the average total length of service was 13.31 years ($\pm$12.63 years; 0.5–40 years), and the average length of service in the urology department was 7.71 years ($\pm$10.29 years; 0.5–37 years). The largest number of nurses had a bachelor's degree (57.7%), and 46% of the respondents did not have additional postgraduate education. Most of the respondents had one job (61%), an average of 1.45 jobs ($\pm$0.6 jobs) and worked in a 12 h shift system (83%). On average, there were 9.56 patients per one nurse in the urology department ($\pm$4.62 patients; 1–30 patients per one nurse). The nurses had an average of 4.23 diseases ($\pm$2.46; 0–11 diseases). Details are included in Table 1.

**Table 1.** Socio-demographic characteristics of the study group.

| Marital Status | Frequency | Percent |
|---|---|---|
| married woman/married | 75 | 58.10% |
| single/single | 45 | 34.90% |
| divorcee/divorcee | 9 | 7.00% |
| Place of residence | | |
| city | 100 | 76.90% |
| village | 30 | 23.10% |
| Education | | |
| medium | 15 | 11.50% |
| higher undergraduate | 75 | 57.70% |
| graduate and above | 40 | 30.80% |
| Work system | | |
| single shift | 15 | 11.50% |
| shift | 115 | 88.50% |
| Satisfaction with the financial situation | | |
| no | 33 | 25.40% |
| on average | 62 | 47.70% |
| yes | 35 | 26.90% |
| Staying on sick leave | | |
| no | 84 | 64.60% |
| yes | 46 | 35.40% |
| I want to change my job | | |
| no | 35 | 26.90% |
| maybe | 55 | 42.30% |
| yes | 40 | 30.80% |

Source: own study.

### 2.4. Data Collection Procedures and Statistical Procedures

The questionnaires were made available to nurses on paper and online in urology departments. Then, the collected data from the questionnaires were placed in an Excel spreadsheet and statistical analysis was performed. The questionnaires with missing data were excluded.

First, the existence of correlations between independent variable predictors was verified in order to eliminate those variables that strongly correlate with other variables. Then, the impact of the other predictors on care rationing and the degree of effectiveness of the independent variables in explaining the dependent variable were tested.

The significance level was $p = 0.05$. Accordingly, results of $p < 0.05$ indicate significant relationships between the variables.

### 2.5. Ethical Procedure

It was explained that participation in the study is voluntary and anonymous. Consent from the nurses and request and receipt of written consent occurred prior to the assessment. Prior to the study, the consent of the Bioethics Committee of the Medical University of Silesia in Katowice (Ethical Number: PCN/CBN/0052/KB/32/22) was obtained. The STROBE (Strengthening the Reporting of Observational studies in Epidemiology) guidelines for observational studies were followed.

## 3. Results

### 3.1. Basic Descriptive Statistics

In the surveyed group of nurses, the average rationing of nursing care obtained 1.11 points (±0.70 points), what means rare rationing, general job satisfaction obtained 5.95 points (±1.92 points) out of 10 points, while the average assessment of the quality of patient care obtained 6.88 points (±1.76 points) out of 10 points. Details are shown in Table 2.

**Table 2.** Descriptive statistics of care rationing.

| PIRNCA | M | SD | Min | Max | Me |
|---|---|---|---|---|---|
| Rationing of nursing care | 1.11 | 0.70 | 0.00 | 2.97 | 1.06 |
| Overall job satisfaction | 5.95 | 1.92 | 1.00 | 10.00 | 6.00 |
| Assessment of the quality of patient care | 6.88 | 1.76 | 3.00 | 10.00 | 7.00 |

M—medium; SD—standard deviation; Min—minimum; Max—maximum; Me—median. Source: own study.

### 3.2. Correlation Analysis

The existence of correlations between independent variable predictors was verified in order to eliminate those variables that strongly correlate with other variables. Details are shown in Table 3.

**Table 3.** Correlation analysis of independent variables (predictors).

| | Age | Postgraduate | Workplaces | Seniority | Illnesses | Operating Mode | Location Financial | Willingness to Change |
|---|---|---|---|---|---|---|---|---|
| Postgraduate | 0.326 *** | — | | | | | | |
| Workplaces | 0.029 | 0.148 | — | | | | | |
| Work experience in the profession | 0.895 *** | 0.409 *** | 0.015 | — | | | | |
| Internship in the urology department | 0.745 *** | 0.398 *** | −0.061 | 0.822 *** | | | | |
| Number of the sick | 0.184 * | 0.124 | 0.280 ** | 0.159 | | | | |
| Number of diseases | 0.021 | 0.010 | 0.094 | 0.100 | — | | | |
| Education | −0.013 | 0.231 ** | 0.093 | −0.045 | −0.014 | | | |
| Operating mode | −0.155 | −0.023 | 0.037 | −0.152 | 0.067 | — | | |
| Financial situation | 0.054 | 0.043 | −0.136 | 0.049 | −0.150 | 0.008 | — | |
| Staying on sick leave | 0.098 | −0.085 | −0.051 | 0.116 | 0.314 *** | −0.085 | −0.105 | |
| Willingness to change | −0.243 ** | −0.007 | 0.079 | −0.165 | 0.263** | 0.175 * | −0.311 *** | — |
| Marital status | −0.452 *** | −0.186 * | 0.174 * | −0.370 *** | 0.034 | 0.103 | −0.018 | 0.201 * |
| Place of residence | −0.084 | −0.103 | −0.001 | −0.139 | −0.023 | 0.084 | 0.139 | −0.126 |

Note: * $p < 0.05$, ** $p < 0.01$, *** $p < 0.001$. Source: own study.

Correlation analysis using the Spearman's test showed that the variables strongly correlating with other independent variables, and thus necessary to be excluded from the analysis, are age, seniority in the nursing profession, seniority, and marital status. Education, in turn, strongly correlated with the number of forms of post-graduate education. In this case, it was decided to remove the education variable and leave the variable form of post-graduate education, because, as verified, the number of forms of post-graduate education appeared to be significant in the regression model used, unlike education. Similarly, the sick leave variable strongly correlated with the variable number of diseases; this variable had a significant impact on the obtained results of the scales used, which resulted in the removal of the sick leave variable. The variable number of jobs, irrelevant for further analysis, correlated with a significant variable number of patients, so the variable number of jobs was removed. A strong correlation between the variables of satisfaction with the financial situation and willingness to change jobs made it necessary to remove the variable

willingness to change jobs from the regression analysis. The removal of these variables was necessary to maintain the methodological correctness, because one of the assumptions regarding the regression model is the independence of observation errors, i.e., the lack of correlation.

### 3.3. Analysis of Variance

Test result F (6.120) = 2.785; $p$ = 0.014 is statistically significant and indicates that sociodemographic characteristics have a statistically significant impact on the rationing of nursing care.

Test result F (6.123) = 7.552; $p$ < 0.001 is statistically significant and indicates that sociodemographic characteristics have a statistically significant impact on overall job satisfaction.

Test result F (6.123) = 1.186; $p$ = 0.318 is not statistically significant and indicates that sociodemographic characteristics do not have a statistically significant impact on the assessment of the quality of patient care.

Details are presented in Table 4.

**Table 4.** Analysis of variance of the influence of sociodemographic characteristics on the rationing of nursing care, job satisfaction and the quality of patient care.

|  | **Model** | **SS** | **df** | **MS** | **f** | **p** |
|---|---|---|---|---|---|---|
| PIRNCA | Regression | 7484 | 6 | 1.247 | 2785 | 0.014 |
|  | Rest | 53,742 | 120 | 0.448 |  |  |
|  | Total | 61.227 | 126 |  |  |  |
| PIRNCA (job satisfaction) | Regression | 128.067 | 6 | 21,345 | 7.552 | <0.001 |
|  | Rest | 347.656 | 123 | 2.826 |  |  |
|  | Total | 475.723 | 129 |  |  |  |
| PIRNCA (quality of patient care) | Regression | 21,766 | 6 | 3.628 | 1.186 | 0.318 |
|  | Rest | 376.264 | 123 | 3.059 |  |  |
|  | Total | 398.031 | 129 |  |  |  |

Note: SS—sum of squares; df—degrees of freedom; MS—mean square; F—test statistics; $p$—statistical significance; PIRNCA–rationing of nursing care. Source: own study.

### 3.4. Regression Analysis

A detailed analysis showed a significant impact of the number of diseases occurring in the subjects on the rationing of nursing care. The regression coefficient was 0.225; t(120) = 2.552; $p$ = 0.012. This means that as the number of diseases increases by one point, the rationing of nursing care increases by 2.225 points. There is a significant impact of the number of diseases on the rationing of nursing care. Details are shown in Table 5.

**Table 5.** Regression analysis of the impact of sociodemographic characteristics on the rationing of nursing care, overall job satisfaction, the quality of patient care.

| Model | PIRNCA | | | | | PIRNCA (General Job Satisfaction) | | | | | PIRNCA (Quality of Patient Care) | | | | |
|---|---|---|---|---|---|---|---|---|---|---|---|---|---|---|---|
|  | b | SE | β | t | p | b | SE | β | t | p | b | SE | β | t | p |
| (Constant) | −0.205 | 0.451 |  | −0.456 | 0.649 | 3.827 | 1.106 |  | 3.459 | 0.001 | 6.533 | 1.151 |  | 5.677 | <0.001 |
| Place of residence | 0.160 | 0.146 | 0.097 | 1.097 | 0.275 | 0.722 | 0.360 | 0.159 | 2.008 | 0.047 * | −0.134 | 0.374 | −0.032 | −0.359 | 0.720 |
| Number of forms of postgraduate education | 0.031 | 0.074 | 0.036 | 0.414 | 0.680 | 0.160 | 0.184 | 0.068 | 0.869 | 0.386 | 0.314 | 0.192 | 0.146 | 1.637 | 0.104 |
| Work system | 0.215 | 0.192 | 0.097 | 1.120 | 0.265 | −0.108 | 0.467 | −0.018 | −0.231 | 0.818 | 0.286 | 0.485 | 0.052 | 0.590 | 0.557 |
| Number of the sick | 0.029 | 0.013 | 0.192 | 2.171 | 0.032 | −0.047 | 0.033 | −0.114 | −1.423 | 0.157 | −0.056 | 0.035 | −0.147 | −1.616 | 0.109 |
| Satisfaction with the financial situation | 0.073 | 0.086 | 0.075 | 0.849 | 0.398 | 1.060 | 0.211 | 0.401 | 5.018 | <0.001 *** | 0.204 | 0.220 | 0.084 | 0.930 | 0.354 |
| Number of diseases | 0.064 | 0.025 | 0.225 | 2.552 | 0.0128 * | −0.084 | 0.062 | −0.108 | −1.367 | 0.174 | −0.031 | 0.064 | −0.044 | −0.487 | 0.627 |

Note: B—non-standardized coefficient; SE—standard error; β—standardized coefficient; t—test statistics; $p$—statistical significance; PIRNCA—rationing of nursing care. * $p$ < 0.05; *** $p$ < 0.001. Source: own study.

A detailed analysis showed a significant impact of satisfaction with the financial situation on overall job satisfaction. The value of the regression coefficient was 0.401; $t(123) = 5.018$; $p < 0.001$. This means that as satisfaction with the situation increases by one point compared to dissatisfaction, there is an increase in overall job satisfaction by 0.401 points. A significant impact of the place of residence on overall job satisfaction was also demonstrated. The regression coefficient was 0.068; $t(123) = 0.869$; $p = 0.047$. This means that with the change in the place of residence by one point, the overall job satisfaction increases by 0.068 points in comparison with the countryside and the city. There is a significant impact of satisfaction with the financial situation and place of residence on overall job satisfaction. Details are presented in Table 5.

A detailed analysis confirmed the lack of a significant impact of sociodemographic variables on the overall assessment of the quality of nursing care. Details are presented in Table 5.

### 3.5. Fit Models

The adjusted $R^2 = 0.078$ means that the model explains 7.8% of the variance in nursing care rationing. The adjusted $R^2 = 0.234$ means that the model explains 23.4% of the variability in the overall job satisfaction. The adjusted $R^2 = 0.09$ means that the model explains 0.9% of the variability in the assessment of the quality of nursing care. Details are presented in Table 6.

**Table 6.** Matching model of the impact of sociodemographic data on rationing of nursing care, overall job satisfaction, the quality of patient care.

| Model | $R^2$ | Corrected $R^2$ | The Standard Error of the Estimate |
|---|---|---|---|
| PIRNCA | 0.122 | 0.078 | 0.66922 |
| PIRNCA (general job satisfaction) | 0.269 | 0.234 | 1.681 |
| PIRNCA (Quality of Patient Care) | 0.055 | 0.009 | 1.749 |

Note: $R^2$—model fit factor; PIRNCA—rationing of nursing care. Source: own study.

## 4. Discussion

In the present study, the nursing care was rationed rarely (1.11 points), the job satisfaction was on an average level (5.95 points) and the assessment of the quality of patient care was also on an average level (6.88 points). An impact of the number of nurses' diseases on the rationing nursing care, impact of satisfaction with the financial situation and the place of residence on overall job satisfaction were shown. The quality of care was not influenced by any of the analyzed factors.

The conducted own research on the rationing of care in urological wards is the first in this field in wards of this specialty; therefore, the results were compared with wards of other specialties, as well as to those obtained in other countries. For the first time, the influence of selected sociodemographic factors (place of residence, postgraduate education, work system, number of patients per nurse, satisfaction with the financial situation, number of diseases of a nurse) on care rationing was analyzed.

Our own study showed that nursing care was rarely rationed (1.11 points). The result of sparse rationing of care is confirmed by other authors' studies. Radosz-Knawa et al. covered 11 medical units and obtained an average care rationing score of 1.94 points, which means that care was also rarely rationed. However, the results of care rationing differed depending on the facility and ranged from 1.39 points (hospital 5) to 2.43 points (hospital 1), with the highest score being "sometimes" for rationing of care [16]. Rare rationing of care was also found among oncology nurses; the researchers obtained an average 1.55 points [27]. Tomaszewska et al. in their research showed that nurses working in surgical wards resulted in 2.72 points, i.e., care was sometimes rationed, and the value was higher than that of nurses from conservative wards (2.08 points) [28]. Kołtuniuk et al. achieved a rationing score of 1.53 points [18]. In their research, Młynarska et al. obtained an average care

rationing score of 0.81 points [17]. Comparing our own study with other studies in Poland, it can be stated that the obtained result is one of the lowest; lower results were only obtained by Młynarska et al. in intensive care units. Comparing our own research with the results of surgical departments, which include urology departments, the result is more than half lower. This indicates that the rationing of care in urology departments differs significantly from surgical departments, and the results from these departments should be separated in future analyses.

Exactly the same rationing result as in our own research was obtained by Maghsoud et al. [29]. Research conducted at the Tertiary Public Hospital in the Philippines showed an average care rationing rate of 1.04 points [30]. In a Swiss study by Shubert et al., respondents rarely rationed care (1.69 points in BERNCA) [14]. Zelenikova et al., in research among nurses from four European countries, showed that the average level of care rationing was the following: Slovakia–1.38 points; Czech Republic—1.28 points; Poland—1.7 points; Croatia—1.76 points [31]. Shubert et al., in the 2009 study, received a care rationing score of 0.82, which means a slightly less often occurring rationing than rarely [32]. The international results mainly mean rare rationing of care as in our own research, and only the study by Shubert et al. showed rationing of care less than rare.

Kołtuniuk et al. showed the influence of the subjects' age, work experience and the number of patients per nurse [18]. Kalánková et al. also showed the impact of sociodemographic factors on care rationing, such as education, specialization, overtime, willingness to leave the profession, expected staff adequacy, job satisfaction and work experience [33]. However, Młynarska et al. showed no relationship between sociodemographic factors (age, marital status, ward where they work, seniority, education) and care rationing [17]. These factors are extremely important in the work of a nurse, but they were not included in our own research, which confirms the lack of previous research in this area.

Młynarska et al. showed that the average job satisfaction was 7.13 points and the average quality of care score was 6.05 points [17]. Equally high results of job satisfaction and quality of patient care were obtained by Kalánková et al. in studies in Slovak intensive care units. The average assessment of the quality of care was 7.94 points, which was defined as a high level, and the overall job satisfaction was 7.24 points, which was considered as the satisfaction of nurses with their work [33]. In own research, lower results were obtained in relation to the predecessors, which means lower overall job satisfaction (5.95 points) and a lower average assessment of the quality of patient care (6.88 points). Authors such as Shubert et al. [34], Ausserhofer et al. [9] and Zhu et al. [35] showed that the study groups were neither dissatisfied nor satisfied with their work, these results are confirmed by other authors: Uchmanowicz et al. [13,36], Jaworski et al. [5]. These results can be related to our own results, which are almost in the middle of the scale of job satisfaction-5.95 points/10 points.

On the other hand, our own research showed that the number of diseases nurses suffer from significantly affects the rationing of care, while job satisfaction is affected by satisfaction with the financial situation and place of residence, and none of the analyzed variables affects the quality of care. There have been no previous studies analyzing the impact of the number of nurses' illnesses on care rationing, therefore it is impossible to refer to other studies, which may be the subject of future research.

Zúñiga et al. showed that nurses had higher quality of care when they rationed care less often [37]. Our research did not show any significant impact of sociodemographic factors on the quality of care. In contrast, care was rarely rationed, with nurses rating the quality of care at 6.88 points, consistent with the findings of Zúñig et al.

Researchers have shown that nurses who are less satisfied with their jobs ration care 2.6 [38] to 3.4 [39] times more often than satisfied nurses. Kalisch and Williams showed that nurses who rationed were less satisfied with their jobs [40]. Zeleníková et al. showed that care rationing was significantly correlated with job satisfaction, assessment of care quality, intention to leave the current job, and perceived staff adequacy [31]. Clark et al. showed that 10.9% of respondents often rationed care, and this percentage increased to 26.5% for

people dissatisfied with their jobs [39]. With regard to our own research, it can be stated that nurses may be exposed to care rationing due to the level of their job satisfaction.

The authors most often mention the following factors as having a significant impact on job satisfaction: the atmosphere at work, physical effort, stability of employment, relationships in the team, workplace equipment, defining the role at work, opportunities to improve qualifications, stress and remuneration [41,42] reduction in staff, increased demand for care, new technologies, a wider range of treatment [5]. These factors did not include any factors related to the nurse's private sphere, such as the number of diseases or place of residence.

Andruszkiewicz et al. showed that the perceived low level of job satisfaction as well as the material situation and assessment of remuneration have a negative impact on the health of nurses. Nurses who assess their remuneration and financial situation as weaker also have lower job satisfaction [42]. Research by Gawęda et al. confirmed that the cause of job dissatisfaction is low pay [43]. Młynarska et al. in intensive care units showed that only 54.67% of the respondents were satisfied with their financial situation. On the other hand, fatigue of nurses had a significant impact on the rationing of care; no socio-demographic factor was determined [17]. These results confirm the results of our own research, i.e., the significant impact of satisfaction with the financial situation on job satisfaction.

Radosz-Knawa et al. showed that the level of care rationing increased with the increase in occupational burnout, while the age of the respondents had an inversely proportional effect on care rationing. The authors explain the differences in the rationing of care between hospitals by the number of patients under the care of nurses, which was in the range of 13.11 (hospital 1)–6.31 (hospital 5) patients. They showed a correlation between the number of adverse events and the assessment of the work environment [16]. Cisek et al. showed that there were 12–24 patients per nurse on duty [44], compared to 6–11 patients in the UK [45]. In our own research, there were 9.56 patients per nurse, which is within the range provided by Radosz-Knawa and in Great Britain, but is lower than in the studies of Cisek et al. Ball et al. showed that nurses caring for 11.7 patients or more increased the risk of rationing care by 66% [45]. With reference to the research by Ball et al., it can be concluded that there is a risk of rationing care in urological wards, but it is not as high as indicated by Ball et al. The Polish Nursing Association has determined that the quality of services, patient and nurse safety is determined by the appropriate nursing staff [46].

Many studies have been conducted on the impact of various factors on care rationing, mainly age, work experience, occupational burnout, education, etc., and they have shown significant correlations. The authors' own research includes factors that have not been studied before, which makes it impossible to compare them, but points to a new path for further research. The main limitations in the study were: (1) limitations in spreading the questionnaire results of restrictions included in Poland connected with SARS-CoV-2; (2) the risk of filling the questionnaires by people who are not nurses or nurses working in the other departments, especially in the case of producing the online version; (3) the lack of the previous studies such as an own study in the urology departments, which causes an absence of points to compare.

## 5. Conclusions

The level of care rationing in urology departments does not differ significantly from care rationing in other departments. Socio-demographic factors do not affect the quality of patient care. Satisfaction with the financial situation affects the nurse's job satisfaction. Even though the urology wards do not differ significantly from other wards, they should constantly control the level of the rationing nursing care and implement the preventive solutions. The nurses should take care of themselves and their self-development. Additionally, the hospitals should consider some solutions to increase the nurses' financial satisfaction and promote prevention of nurses' health. The most positive finding is the lack of impact on the quality of patient care, which shows nurses' professionalism. The subject of the impact of the sociodemographic factors on the rationing nursing care is not

well known and it is really wide, which implies that more research is needed in the future, especially in the urology wards.

**Author Contributions:** Conceptualization, K.J. and A.M.; methodology, K.J. and A.M.; software, A.M.; validation, K.J. and A.M.; formal analysis, K.J. and A.M.; investigation, K.J. and A.M.; resources, K.J.; data curation, K.J. and A.M.; writing—original draft preparation, K.J.; writing—review and editing, A.M.; visualization, K.J. and A.M.; supervision, K.J. and A.M.; project administration, K.J. and A.M.; funding acquisition, K.J. and A.M. All authors have read and agreed to the published version of the manuscript.

**Funding:** This research received no external funding.

**Institutional Review Board Statement:** The study was conducted in accordance with the Declaration of Helsinki, and approved by the Bioethics Committee of the Medical University of Silesia in Katowice (Ethical Number: PCN/CBN/0052/KB/32/22) for studies involving humans.

**Informed Consent Statement:** Informed consent was obtained from all subjects involved in the study.

**Data Availability Statement:** The datasets generated during and/or analyzed during the current study are available from the corresponding author on reasonable request.

**Conflicts of Interest:** The authors declare no conflict of interest.

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
