# Peer review of "The Impact of Sociodemographic Factors on the Rationing of Nursing Care in Urology Wards"

_nursrep, doi:10.3390/nursrep13010051_

Round 1

Reviewer 1 Report

Thank you for the opportunity to read and review the article "The impact of sociodemographic factors on the rationing of nursing care in urology wards"

However, after having learned the article and seriously considered its context I'm afraid I find it not only disorganized and inconsistent but also that its main final conclusion-" bottom line" is rather unclear and unconvincing.

My reasons:

1. In the beginning the article deals with the influence of factors and physical and mental health of the nurse.  Obviously, when an article begins in such a manner the reader expect that it would deal with the connection with those factors and the nurse rationing process. But this article deals only with the quality of care and the researchers do not explain the connection between rationing and the nurses health.

 2. The researchers point out that risk factors include atmosphere in the ward, a large number of patients under the care  of one nurse, shift work, variety of tasks, technical resources, and work should be eliminated to minimize the risk of decrease of quality of care.  But these risk factors cannot possibly be eliminated from reality and or disregarded. Therefore, this statement is inaccurate. In addition, the researchers do not explain the means and ways that nurses have to handle the situations.

3.  Regarding the influence of sociodemographic factors on rationing , it is not at all clear why these socio-demographic factors were chosen, and there is no explanation available in the introduction section as to having chosen those factors from the literature.

The researches confused job satisfaction with rationing and quality of care. This confusion would make the reader unsure whether he is reading about rationing or job satisfaction.

In the end of the article , the researchers conclude- "no connection was found, but a connection was found between the economic situation and job satisfaction"

 Again, the conclusion does not correspond at all with the topic of the article, and there are no practical recommendations from the literature or on behalf of the researchers that deal with their conclusion. 

After a thorough reading of the article, I would advise rejecting the article.

Author Response

Dear Reviewer,

thank you for your critical review. I have tried reply to your reviews. The details are included in the attached file.

Yours faithfully,

Katarzyna Jarosz

Reviewer 2 Report

1. What are the full terms for the BERNCA measure and the PIRNCA tool? In the first statement, please give the full term.

2. The researcher explained why the subjects of this study were urology nurses as follows. 'Urology tends to be overlooked in many studies or automatically integrated with surgery.' But is this the subjective guess of the researcher? If not, please add a reference. This is a very important amendment request due to the needs of this study.

3. In the discussion, it was said that the result of this study received the lowest distribution score compared to other studies. If so, an analysis of these findings should be added.

4. It was an interesting research topic, but in the area of discussion, research significance should be added through comparative analysis with other previous research results.

Author Response

Dear Reviewer,

Thank you for your valueable review. The details are included in the attached file.

Yours faithfully,

Katarzyna Jarosz

Reviewer 3 Report

Thank you for allowing me to review this manuscript. This manuscript entitled "The impact of sociodemographic factors on the rationing of nursing care in urology wards". The aim of this study was to examine the impact of sociodemographic factors (place of residence, satisfaction with financial situation, number of forms of postgraduate education, work system, number of patients per nurse, number of illnesses) on care rationing, job satisfaction, and quality of nursing care.

It is an interesting project, with a very current theme, although it has several limitations that make it susceptible to publication in this magazine. These limitations are detailed below:

- The introduction is well structured, contextualizing the theoretical framework of the study in a correct and orderly manner. However, I would recommend indicating more clearly the relevance and topicality of the subject of study.

- In the material and methods section it is described in detail. However, it would be important to note whether the sample is representative and how it was calculated. Also, when the applied questionnaires are addressed, it would be necessary to specify the data on validity and reliability.

- The results are presented in a clear and orderly manner. Also, an interpretation of them is reflected. However, there are too many tables. We recommend trying to reduce the number of them or merge.

- In relation to the discussion, the results are discussed in an orderly manner in the manuscript. However, it would be necessary to include the limitations found during the preparation of the manuscript.

- The conclusions are clear and precise. However, I consider it necessary to emphasize the importance of the study that some issues are reviewed. First, I recommend the implication of the conclusions in clinical nursing practice. Also, consider it necessary to include a line for the future that emphasizes the importance of continuing to work on this topic of study.

good job

Author Response

Dear Reviewer,

Thank you very much for your advices. The details are included in the attached file.

Yours faithfully, 

Katarzyna Jarosz

Round 2
